# AdaFuse: Adaptive Multimodal Fusion for Lung Cancer Risk Prediction via Reinforcement Learning

**Chongyu Qu**[1]                   CHONGYU.QU@VANDERBILT.EDU
**Zhengyi Lu**[1]                   ZHENGYI.LU@VANDERBILT.EDU
**Yuxiang Lai**[3]                   YUXIANGLAI117@GMAIL.COM
**Thomas Z. Li**[1]                  THOMAS.Z.LI@VANDERBILT.EDU
**Junchao Zhu**[1]                  JUNCHAO.ZHU@VANDERBILT.EDU
**Junlin Guo**[1]                   JUNLIN.GUO@VANDERBILT.EDU
**Juming Xiong**[1]                 JUMING.XIONG@VANDERBILT.EDU
**Yanfan Zhu**[1]                  YANFAN.ZHU@VANDERBILT.EDU
**Yuechen Yang**[1]                YUECHEN.YANG@VANDERBILT.EDU
**Allen J. Luna**[1,2]                ALLEN.J.LUNA@VANDERBILT.EDU
**Kim L. Sandler**[2]                 KIM.SANDLER@VUMC.ORG
**Bennett A. Landman**[1,2]         BENNETT.LANDMAN@VANDERBILT.EDU
**Yuankai Huo**[1]                 YUANKAI.HUO@VANDERBILT.EDU

[1] *Vanderbilt University, Nashville, TN, USA, 37215*

[2] *Vanderbilt University Medical Center, Nashville, TN, USA, 37232*

[3] *Emory University Atlanta, GA, USA, 30322*

**Editors:** Accepted for publication at MIDL 2026

## Abstract

Multimodal fusion has emerged as a promising paradigm for disease diagnosis and prognosis, integrating complementary information from heterogeneous data sources such as medical images, clinical records, and radiology reports. However, existing fusion methods process all available modalities through the network, either treating them equally or learning to assign different contribution weights, leaving a fundamental question unaddressed: **for a given patient, should certain modalities be used at all?** We present AdaFuse, an adaptive multimodal fusion framework that leverages reinforcement learning (RL) to learn patient-specific modality selection and fusion strategies for lung cancer risk prediction. AdaFuse formulates multimodal fusion as a sequential decision process, where the policy network iteratively decides whether to incorporate an additional modality or proceed to prediction based on the information already acquired. This sequential formulation enables the model to condition each selection on previously observed modalities and terminate early when sufficient information is available, rather than committing to a fixed subset upfront. We evaluate AdaFuse on the National Lung Screening Trial (NLST) dataset. Experimental results demonstrate that AdaFuse achieves the highest AUC (0.762) compared to the best single-modality baseline (0.732), the best fixed fusion strategy (0.759), and adaptive baselines including DynMM (0.754) and MoE (0.742), while using fewer FLOPs than all triple-modality methods. Our work demonstrates the potential of reinforcement learning for personalized multimodal fusion in medical imaging, representing a shift from uniform fusion strategies toward adaptive diagnostic pipelines that learn when to consult additional modalities and when existing information suffices for accurate prediction. Code is publicly available at: https://github.com/hrlblab/adafuse

**Keywords:** Reinforcement Learning, Multimodal Fusion, Risk Prediction

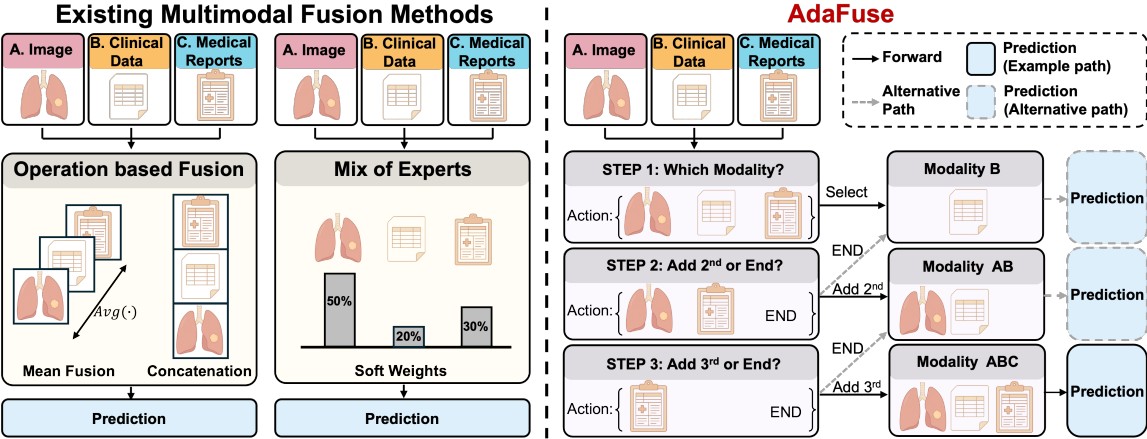

Figure 1: **Comparison of multimodal fusion paradigms.** Existing multimodal fusion methods (left two) process all modalities uniformly: operation-based fusion applies fixed combination rules, while Mixture-of-Experts learns soft weights but still requires all inputs. AdaFuse (right) makes sequential discrete decisions to select patient-specific modality subsets, with the flexibility to entirely exclude uninformative modalities from computation.

## 1. Introduction

Lung cancer remains the leading cause of cancer-related mortality worldwide (Rivera et al., 2013), with early detection being critical for improving patient outcomes (Blandin Knight et al., 2017). Low-dose computed tomography (LDCT) screening has demonstrated significant potential in reducing lung cancer mortality (Bonney et al., 2022; Ostrowski et al., 2018), as evidenced by the National Lung Screening Trial (NLST) (Team, 2011). Beyond imaging, clinical variables such as age, smoking history, and family history provide complementary risk factors (Tammemägi et al., 2013), while radiology reports (Hans Vitzthum von Eckstaedt et al., 2020) capture expert observations and contextual information. The integration of these heterogeneous data sources through multimodal fusion (Cui et al., 2022; Liu et al., 2023) has emerged as a promising direction for more accurate and comprehensive risk prediction

Multimodal fusion methods have evolved considerably over the past years. Early approaches relied on simple operations such as concatenation (Mobadersany et al., 2018; Yap et al., 2018) or mean pooling (Cheerla and Gevaert, 2019; Ghosal et al., 2021) to combine features from different modalities. Tensor-based methods employing Kronecker products (Chen et al., 2020; Wang et al., 2021) were later introduced to capture higher-order feature interactions, though at the cost of increased computational complexity. To address this, low-rank factorization techniques (Liu et al., 2018; Sahay et al., 2020) have been proposed to reduce the parameter overhead while preserving expressive power. Attention mechanisms (Zhu et al., 2020; Lu et al., 2023) have gained popularity for their ability to learn cross-modal interactions and dynamically weight modality contributions. More recently, Mixture-of-Experts (MoE) architectures (Cao et al., 2023; Han et al., 2024) route inputs

through specialized sub-networks, providing input-dependent feature processing. Despite their architectural differences, all these methods process every available modality through the network, either treating them equally or learning to assign different contribution weights (i.e., soft selection), as illustrated in Figure 1 (left). This leaves a fundamental question unaddressed: **for a given patient, should certain modalities be used at all?**

In clinical practice, the diagnostic value of each modality varies across individuals (Acosta et al., 2022; Huang et al., 2020). Some patients may benefit from multimodal integration, while for others, a single modality suffices or additional modalities introduce noise rather than complementary information. Existing methods cannot entirely exclude uninformative modalities from computation; instead, they uniformly process all inputs regardless of their utility for individual patients. This not only incurs unnecessary computational costs but also limits the model's ability to truly personalize the diagnostic pipeline.

To address this limitation, we propose **AdaFuse**, an adaptive multimodal fusion framework that formulates modality selection as a sequential decision process. As shown in Figure 1 (right), at each step, a policy network decides whether to incorporate an additional modality or to proceed to prediction with the current selection. Unlike existing approaches that process all modalities and learn to weight their contributions, AdaFuse makes discrete decisions to select or skip each modality, providing the flexibility to use different modality combinations for different patients. This formulation naturally mirrors clinical practice, where physicians selectively order diagnostic tests based on patient-specific factors rather than exhaustively acquiring all available data (Winslow et al., 1988; Ball et al., 2015).

Our contributions are summarized as follows:

1. We propose AdaFuse, a reinforcement learning framework that formulates multimodal fusion as a sequential decision process, enabling patient-specific modality selection for lung cancer risk prediction.

2. We conduct comprehensive experiments on the NLST dataset with three modalities, comparing against single-modality baselines and various fusion strategies including concatenation, mean pooling, tensor fusion, and MoE.

3. We provide detailed analysis of learned fusion policies, offering insights into how the model adapts its modality selection across different patient subgroups and demonstrating the potential of adaptive fusion for personalized medical diagnosis.

## 2. Method

We present AdaFuse, an adaptive multimodal fusion framework that learns patient-specific modality selection strategies through reinforcement learning. The key insight is that different patients may benefit from different modality combinations: some patients may require comprehensive multimodal integration, while others may achieve accurate predictions with fewer modalities. Rather than applying a fixed fusion strategy uniformly, AdaFuse formulates modality selection as a sequential decision process, where a policy network learns to identify the optimal modality subset for each patient.

Figure 2 illustrates the overall architecture. Given a patient with three available modalities, AdaFuse first encodes each modality into a compact representation. A policy network

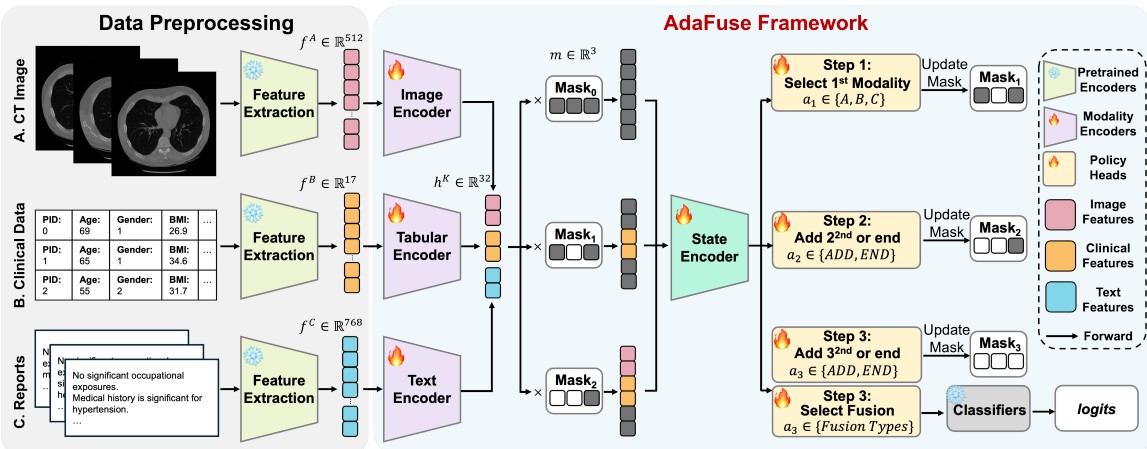

Figure 2: **Overview of the AdaFuse framework.** *Data preprocessing* extracts features from three modalities (CT images, clinical variables, and text reports) using pretrained encoders; details are provided in Section 3.1. The **AdaFuse framework** consists of three components: (1) *Modality encoders* project each input feature to a shared 32-dimensional representation, with a binary mask $m \in \{0, 1\}^3$ tracking selected modalities.(2) *State encoder* concatenates the masked features with the selection mask and maps them to a 64-dimensional state vector that captures the current selection status. (3) *Policy heads* make sequential decisions: Step 1 selects the primary modality from $\{A, B, C\}$ ; Step 2 decides whether to add a second modality or proceed to prediction; Step 3 optionally incorporates the third modality and selects a fusion strategy from $\{Concat, Mean, Tensor\}$. After each selection, the corresponding mask entry is updated from 0 to 1, and the state encoder recomputes the state representation for the next decision. The selected modality combination is passed to the corresponding pretrained classifier among 15 fusion classifiers covering all valid modality-fusion combinations.

then makes sequential decisions: (1) selecting the primary modality, (2) deciding whether to add a second modality, and (3) optionally incorporating the third modality along with a fusion strategy. The selected modalities are fused and passed to a classifier for prediction. The entire framework is trained end-to-end using a combination of supervised learning and policy gradient optimization.

## 2.1. Preliminaries and Notations

We consider a multimodal learning setting with three modalities: CT images, clinical variables, and radiology text reports. For each patient, we denote the raw input features as $f^A \in \mathbb{R}^{512}$ for CT image features, $f^B \in \mathbb{R}^{17}$ for clinical variables, and $f^C \in \mathbb{R}^{768}$ for text embeddings. Each modality is processed by a dedicated encoder $E^A$, $E^B$, $E^C$ to produce encoded representations $h^A = E^A(f^A)$, $h^B = E^B(f^B)$, $h^C = E^C(f^C)$, where all encoded features share a common dimension $d = 32$. The modality selection process is tracked by

a binary mask $\mathbf{m} = [m^A, m^B, m^C]^\top \in \{0, 1\}^3$, where $m^i = 1$ indicates that modality $i$ has been selected.

## 2.2. Sequential Modality Selection

We formulate adaptive modality selection as a Markov Decision Process (MDP) (Bellman, 1957; Puterman, 2014), where the policy learns to sequentially construct the optimal modality subset for each patient.

**State Representation.** At each decision step $t$, the state $s_t$ captures the information from currently selected modalities. We compute the state by concatenating the encoded features weighted by their selection status, along with the mask itself:

$$s_t = g_\theta \left( [h^A \odot m^A; h^B \odot m^B; h^C \odot m^C; \mathbf{m}] \right) \tag{1}$$

Here $\odot$ denotes element-wise multiplication that zeros out unselected modalities, $[\cdot; \cdot]$ denotes concatenation, and $g_\theta$ is a two-layer MLP that maps the $(3d + 3)$-dimensional input to a 64-dimensional state vector. This formulation allows the state encoder to distinguish between "modality not yet selected" (zeroed features, $m^i = 0$) and "modality selected but potentially uninformative" (non-zero features, $m^i = 1$).

**Action Space.** The decision process unfolds over at most three steps, with the action space adapting based on previous selections:

- *Step 1*: The policy selects one modality from $\{A, B, C\}$ as the primary modality.

- *Step 2*: The policy decides whether to stop with the current selection, or to add one of the two remaining modalities.

- *Step 3*: If two modalities have been selected, the policy decides whether to stop or incorporate the third modality, and selects a fusion strategy from {concatenation, mean, tensor}.

This sequential formulation naturally captures the hierarchical nature of modality selection: the primary modality anchors the decision, and subsequent choices refine the combination based on the accumulated information.

## 2.3. Policy Network Architecture

The policy network builds upon the modality encoders, augmented with a state encoder and step-specific decision heads.

**State Encoder.** The state encoder $g_\theta$ is a two-layer MLP with ReLU activations that takes the concatenation of masked modality features and the selection mask as input, producing a 64-dimensional state vector for downstream policy decisions.

**Policy Heads.** Each decision step has dedicated heads that output action logits $\mathbf{l}_t$, from which we sample actions via $\pi(a_t|s_t) = \text{softmax}(\mathbf{l}_t/\tau)$. The temperature $\tau$ is annealed from $\tau_{\text{init}}$ to $\tau_{\text{final}}$ over training to transition from exploration to exploitation; during inference, we use greedy decoding.

**Fusion Classifiers.** We maintain 15 fusion classifiers corresponding to all valid modality-fusion combinations (3 single-modality, 9 dual-modality for 3 pairs × 3 fusion types, and 3 triple-modality for 3 fusion types), and invoke the appropriate classifier based on the policy's selection.

## 2.4. Learning Objective

**Reward Design.** After the policy completes its sequential decisions and produces a prediction $\hat{p}$ for a patient with label $y$, we compute a reward signal based on two components:

$$r = r_{\mathrm{BCE}} + \lambda_{\mathrm{auc}} \cdot r_{\mathrm{auc}} \tag{2}$$

The first term $r_{\mathrm{BCE}} = y \log \hat{p} + (1 - y) \log(1 - \hat{p})$ is the negative binary cross-entropy, which provides a continuous signal based on prediction confidence. The second term $r_{\mathrm{auc}}$ is a mini-batch AUC reward: for each sample, we compute how well its prediction ranks relative to samples of the opposite class, normalized to $[-1, 1]$. Unlike a fixed-threshold indicator (e.g., $\hat{p} > 0.5$), this formulation is well-suited for imbalanced settings like lung cancer prediction where positive probabilities are typically low.

**Policy Gradient Optimization.** We adopt REINFORCE (Williams, 1992; Hu et al., 2023) over more complex algorithms (e.g., PPO (Schulman et al., 2017), GRPO (Shao et al., 2024)) due to the simplicity of our decision process: with at most three steps and a small discrete action space, the variance reduction from the batch-mean baseline and the auxiliary supervised loss provide sufficient stability. For a trajectory $\boldsymbol{\tau} = (a_1, a_2, \ldots)$ of sequential actions with log-probability $\log \pi_\theta(\boldsymbol{\tau}) = \sum_t \log \pi_\theta(a_t | s_t)$, the policy gradient loss is $\mathcal{L}_{\mathrm{PG}} = -\log \pi_\theta(\boldsymbol{\tau}) \cdot (r - \bar{r})$, where $\bar{r}$ is the mean reward over the mini-batch. The total training objective combines this with an entropy regularization term to encourage exploration:

$$\mathcal{L} = \mathcal{L}_{\mathrm{PG}} - \lambda_{\mathrm{ent}} \sum_t H(\pi_\theta(\cdot | s_t)) + \lambda_{\mathrm{sup}} \mathcal{L}_{\mathrm{BCE}} \tag{3}$$

where the final term is a supervised cross-entropy loss that directly optimizes the classifier, providing stable gradients independent of the stochastic policy. To accelerate convergence, we initialize the encoders and classifiers from pre-trained baseline models and use separate learning rates for different components.

## 3. Experiments

We conduct comprehensive experiments to evaluate AdaFuse on the lung cancer risk prediction task. Section 3.1 describes the dataset and feature extraction pipeline. Section 3.2 introduces the baseline fusion strategies. Section 3.3 compares AdaFuse against baseline methods. Section 3.4 presents ablation studies on training configurations and learning objectives. Section 3.5 presents external validation on an independent cohort.

### 3.1. Dataset

We evaluate on the National Lung Screening Trial (NLST) dataset (Team, 2011), a large-scale multi-center study of low-dose CT screening for lung cancer. To ensure fair evaluation,

we use the held-out test set from Ardila et al. (Ardila et al., 2019) that was not seen during Sybil model (Mikhael et al., 2023) training, since our CT image features are extracted using the pre-trained Sybil encoder. The dataset contains 1,847 patients for training and 462 patients for testing, with lung cancer prevalence of 6.44% and 6.06% respectively. We use a binary classification task where the label indicates whether a participant was diagnosed with lung cancer at any point during the NLST follow-up period (up to 6 years), which aligns with the clinical objective of identifying high-risk individuals for continued surveillance.

**Feature Extraction.** For each patient, we extract three modalities: (1) *CT image features* ($f^A \in \mathbb{R}^{512}$): extracted from the Sybil model, a state-of-the-art lung cancer risk prediction network trained on NLST; (2) *Clinical variables* ($f^B \in \mathbb{R}^{17}$): risk factors from the PLCO$_{\text{m2012}}$ model (Tammemägi et al., 2013) including age, smoking history, BMI, and family history; (3) *Text embeddings* ($f^C \in \mathbb{R}^{768}$): we generate synthetic radiology reports from structured clinical variables covering occupational exposures (e.g., asbestos, chemical work, coal mining), medical history (e.g., diabetes, heart disease, hypertension), and secondhand smoke exposure, then extract embeddings using CORe (Yang et al., 2020), a BERT-based model (Lee et al., 2020) pre-trained on chest radiograph reports.

### 3.2. Baseline Fusion Strategies

We compare AdaFuse against 15 fixed fusion baselines covering all valid modality combinations, plus two adaptive baselines.

**Single-Modality Baselines.** We train three single-modality models ($A$, $B$, $C$) using only CT, clinical, or text features respectively.

**Operation-Based Fusion.** For multi-modality combinations, we evaluate three fusion operations: (1) *Concatenation*: features are concatenated along the channel dimension; (2) *Mean*: features are averaged element-wise after projection to a common dimension; (3) *Tensor*: features are fused via Kronecker product following the Tensor Fusion Network formulation (Zadeh et al., 2017). This yields 9 dual-modality models ($AB$, $AC$, $BC \times 3$ fusion types) and 3 triple-modality models ($ABC \times 3$ fusion types).

**Mixture-of-Experts (MoE).** We implement an MoE baseline with 15 expert classifiers covering all valid modality-fusion combinations (3 single-modality + 9 dual-modality + 3 triple-modality). The gating network is a 2-layer MLP (96→64→64→15) that takes concatenated encoded features from all modalities as input and outputs soft

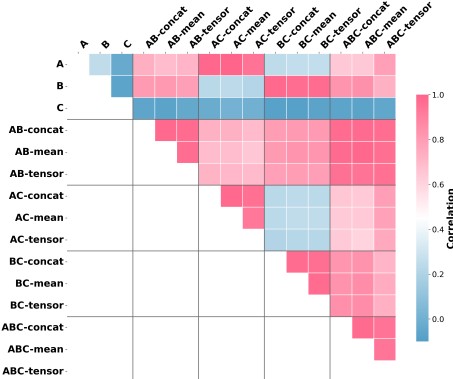

Figure 3: **Prediction correlation across baseline models.** Models containing CT features exhibit high mutual correlation, while text-only predictions show near-zero correlation with others.

weights over experts. Expert models are pre-trained on respective modality combinations, then frozen during gating training. Unlike AdaFuse which makes discrete sequential selection decisions, MoE processes all experts simultaneously and learns continuous soft weights.

**Dynamic Multimodal Fusion (DynMM).** We implement DynMM (Xue and Marculescu, 2023), which uses Gumbel-Softmax gating to make parallel modality selection decisions. Unlike AdaFuse's sequential formulation, DynMM commits to all selection decisions simultaneously based on the initial input features.

### 3.3. Comparison with Baselines

Table 1: **Test AUC and computational cost comparison on NLST.** We compare AdaFuse against single-modality baselines ($A$: CT image, $B$: clinical variables, $C$: text reports), operation-based fusion baselines with three fusion strategies (concatenation, mean pooling, tensor fusion), and adaptive baselines including MoE and DynMM. Background colors indicate modality count: single-modality, dual-modality, and triple-modality. MFLOPs denotes million floating-point operations. The best AUC within each category and the overall best are shown in **bold**.

| Method | AUC | MFLOPs |
|---|---|---|
| $A$ (CT) | **0.732** | 0.543 |
| $B$ (Clinical) | 0.662 | 0.017 |
| $C$ (Text) | 0.576 | 1.067 |
| $AB$-concat | **0.758** | 0.559 |
| $AB$-mean | 0.755 | 0.557 |
| $AB$-tensor | 0.735 | 0.433 |
| $AC$-concat | 0.733 | 1.610 |
| $AC$-mean | 0.745 | 1.608 |
| $AC$-tensor | 0.739 | 1.477 |
| $BC$-concat | 0.661 | 1.084 |
| $BC$-mean | 0.678 | 1.082 |
| $BC$-tensor | 0.685 | 1.088 |
| $ABC$-concat | 0.735 | 1.626 |
| $ABC$-mean | 0.748 | 1.622 |
| $ABC$-tensor | **0.759** | 1.790 |
| MoE | 0.742 | 3.492 |
| DynMM | 0.754 | 1.635 |
| **AdaFuse (Ours)** | **0.762** | 1.164 |

Table 1 presents the test AUC of all methods. AdaFuse achieves the highest AUC (0.762), compared to the best fixed fusion baseline $ABC$-tensor (0.759) and adaptive baselines including DynMM (0.754) and MoE (0.742). Several observations emerge from these results.

**CT features dominate prediction.** The single-modality CT model ($A$, 0.732) already achieves competitive performance, surpassing many multi-modality fusion methods. This aligns with the fact that CT image features are extracted from Sybil, a model specifically trained for lung cancer risk prediction.

**Text reports provide limited information.** The text-only model ($C$, 0.576) performs near random, and combinations involving text without CT ($BC$) yield consistently lower AUC. This is expected since our text reports are synthetically generated from structured clinical variables, which limits their informativeness compared to real radiology reports.

**Naive fusion can hurt performance.** Adding modalities does not guarantee improvement. For instance, *ABC*-concat (0.735) achieves lower AUC than *AB*-concat (0.758), suggesting that indiscriminately incorporating all modalities can introduce noise rather than complementary information.

**Sequential selection benefits from conditioning on observed modalities.** Ada-Fuse (0.762) achieves higher AUC than both DynMM (0.754), which uses parallel Gumbel-Softmax gating, and MoE (0.742), which learns soft modality weights. We chose sequential over parallel selection because it reflects how physicians interpret initial test results before deciding whether to order additional diagnostics. Sequential formulation allows each decision to be conditioned on previously observed modalities, whereas parallel approaches must commit to all selection decisions simultaneously based solely on the initial input features.

**Adaptive selection reduces computational cost.** Among adaptive methods, Ada-Fuse (1.164 MFLOPs) uses 29% fewer FLOPs than DynMM (1.635 MFLOPs) and 67% fewer FLOPs than MoE (3.492 MFLOPs). AdaFuse also uses fewer FLOPs than all triple-modality fixed fusion methods (1.622–1.790 MFLOPs). By learning to skip uninformative modalities for individual patients, AdaFuse achieves the highest AUC while maintaining lower computational cost.

**AdaFuse learns to filter uninformative modalities.** Figure 3 shows the prediction correlation matrix across baseline models. We observe high correlation among models that include CT features, while text-only predictions show near-zero correlation with others. AdaFuse learns to leverage this structure: it predominantly selects CT-based combinations while adaptively incorporating clinical variables when beneficial, effectively filtering out the less informative text modality for most patients. Detailed analysis of the learned policy behavior is provided in Appendix Figure 5.

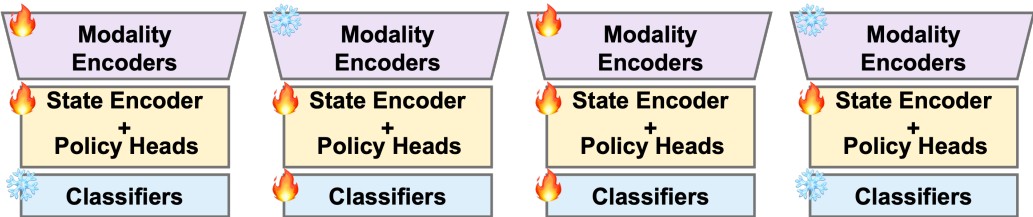

Figure 4: **Ablation study on training configurations.** From left to right: (1) freezing classifiers while training modality encoders; (2) freezing modality encoders while training classifiers; (3) training both components; (4) freezing both components. The flame icon indicates which components receive gradients during RL training. Quantitative results are provided in Table 2.

### 3.4. Ablation Studies

We conduct ablation studies to analyze the contribution of each component in AdaFuse. Section 3.4.1 investigates training configurations, and Section 3.4.2 examines the learning objective design.

### 3.4.1. Training Configuration

Figure 4 illustrates the four training configurations, and Table 2 summarizes the results for freezing versus training encoders and classifiers during RL training.

Table 2: **Ablation study on training configurations.** $\Delta$ denotes relative change compared to the best configuration.

| Encoder | Classifier | Test AUC | $\Delta$ |
|---------|-----------|----------|----------|
| Train | Freeze | **0.762** | – |
| Freeze | Train | 0.722 | -5.25% |
| Train | Train | 0.691 | -9.32% |
| Freeze | Freeze | 0.674 | -11.55% |

**Freezing classifiers yields the best performance.** As shown in Table 2, training classifiers with policy gradients degrades AUC from 0.762 to 0.722 ($-5.25\%$), and jointly training both components further drops to 0.691 ($-9.32\%$). This is because unfrozen classifiers only receive gradients when selected, causing undertrained combinations to produce unreliable rewards that further discourage their selection. Freezing pretrained classifiers provides stable reward signals and allows the policy to focus on selection without shifting decision boundaries.

**Training encoders is essential for policy learning.** The freeze-both configuration yields the worst performance (0.674, $-11.55\%$), indicating that encoder adaptation is necessary. Unlike classifiers that provide fixed decision boundaries, encoders must learn representations that help the policy distinguish when each modality combination is beneficial for a given patient.

### 3.4.2. Learning Objective

We ablate the reward function (Eq. 2) and loss function (Eq. 3) to understand the contribution of each component. Table 3 presents the results.

Table 3: **Ablation study on learning objective.** We separately vary the loss composition (top) and reward design (bottom) while keeping the other fixed. $\mathcal{L}_{\mathrm{PG}}$: policy gradient loss, $\mathcal{H}$: entropy regularization, $\mathcal{L}_{\mathrm{sup}}$: supervised cross-entropy loss. $\Delta$ denotes relative change compared to the best configuration.

| | Configuration | Test AUC | $\Delta$ |
|---|---|---|---|
| **Best** | $\mathcal{L}_{\mathrm{PG}} + 0.1\mathcal{H} + 0.3\mathcal{L}_{\mathrm{sup}}, \quad r = 0.7r_{\mathrm{BCE}} + 0.3r_{\mathrm{AUC}}$ | **0.762** | – |
| Same Reward | $\mathcal{L}_{\mathrm{PG}} + 0.1\mathcal{H}$ (no supervision) | 0.696 | -8.7% |
| | $\mathcal{L}_{\mathrm{PG}} + 0.3\mathcal{L}_{\mathrm{sup}}$ (no entropy) | 0.610 | -20.0% |
| | $\mathcal{L}_{\mathrm{PG}}$ (policy gradient only) | 0.689 | -9.6% |
| | $\mathcal{L}_{\mathrm{PG}} + 1.0\mathcal{L}_{\mathrm{sup}}$ (over-weighted supervision) | 0.641 | -15.9% |
| Same Loss | $r = r_{\mathrm{AUC}}$ (AUC reward only) | 0.647 | -15.1% |
| | $r = r_{\mathrm{BCE}}$ (BCE reward only) | 0.637 | -16.4% |

**Both entropy and supervision are necessary for stable training.** Under the same reward, removing entropy regularization causes the largest performance drop from 0.762 to 0.610 ($-20\%$), as the policy converges prematurely to a narrow set of modality combinations before exploring alternatives. Removing supervision degrades AUC to 0.696, since the supervised loss provides stable gradients independent of stochastic action sampling. However, over-weighting supervision ($\lambda_{\text{sup}} = 1.0$) drops AUC to 0.641 by diminishing the influence of the reward signal.

**Mixed reward outperforms single-objective alternatives.** Under the same loss, using either BCE or AUC reward alone degrades AUC to 0.637 and 0.647, respectively. BCE provides dense per-sample feedback but can be misleading in imbalanced settings. AUC rewards correct ranking and is essential when positive prevalence is only 6%, but provides sparser feedback. The combination leverages the stability of BCE and the ranking awareness of AUC.

### 3.5. External Validation

To assess generalizability, we evaluate AdaFuse on the Vanderbilt Lung Screening Program (VLSP) dataset, a private external cohort with 858 patients (2.8% positive rate). VLSP contains only two modalities (CT images and clinical variables) as it lacks the comprehensive variables required to generate synthetic text reports. Table 4 presents the results.

Table 4: **External validation on VLSP dataset.** VLSP is an independent cohort from the Vanderbilt Lung Screening Program with 858 patients. Only modalities A (CT) and B (clinical) are available.

| Method | VLSP AUC |
|---|---|
| $A$ (CT) | 0.771 |
| $B$ (Clinical) | 0.471 |
| $AB$-concat | 0.725 |
| $AB$-mean | 0.706 |
| $AB$-tensor | 0.590 |
| AdaFuse | **0.749** |

CT features generalize well to VLSP (0.771 AUC), while clinical features show degraded performance (0.471 AUC). This negatively impacts all fixed dual-modality fusion methods, which are forced to incorporate clinical information regardless of its quality. AdaFuse achieves 0.749 AUC, higher than all dual-modality fusion baselines, demonstrating its ability to adaptively filter less informative modalities under distribution shift.

## 4. Conclusion

We presented AdaFuse, an adaptive multimodal fusion framework that addresses a fundamental question in multimodal learning: for a given patient, should certain modalities be used at all? By formulating modality selection as a sequential decision process, AdaFuse learns patient-specific fusion strategies with the flexibility to entirely exclude uninformative modalities rather than processing all inputs uniformly.

Our experiments on NLST reveal that the learned policy predominantly selects CT-based combinations while adaptively incorporating clinical variables, effectively filtering out the less informative text modality for most patients. This adaptive behavior outperforms fixed fusion strategies while demonstrating that uniform fusion is not always optimal. Ablation studies further show that freezing pretrained classifiers is essential for stable policy learning by providing reliable reward signals.

**Potential Limitations.** Our evaluation is limited to three modalities on a single dataset, where text is synthetically generated. Future work could extend AdaFuse to real clinical text and more modalities, and investigate when multimodal integration provides the most benefit across patient subgroups.

## Acknowledgements

This research was supported by the National Institutes of Health (NIH) through grants F30CA275020, 2U01CA152662, R01CA253923 (Landman & Maldonado), R01CA275015 (Maldonado & Lenburg), U01CA152662 (Grogan), U01CA196405 (Maldonado), and P30CA068485-29S1, as well as the National Science Foundation (NSF) through CAREER 1452485 and grant 2040462. Additional support was provided by the Vanderbilt Institute for Surgery and Engineering through T32EB021937-07, the Vanderbilt Institute for Clinical and Translational Research via UL1TR002243-06, the Pierre Massion Directorship in Pulmonary Medicine, and the American College of Radiology Fund for Collaborative Research in Imaging (FCRI) Grant. This manuscript was polished using AI-assisted editing (ChatGPT) with a "rephrase" prompt; no scientific content was generated or altered.

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

# Appendix

## Appendix A. Implementation Details

We provide implementation details for reproducibility. All experiments are conducted on NVIDIA Ampere GPUs.

**Baseline Fusion Models.** We train 15 fusion classifiers using AdamW optimizer (lr=$1 \times 10^{-3}$, weight decay=$1 \times 10^{-4}$), batch size 32, and early stopping with patience of 15 epochs. Each modality encoder projects input features to a 32-dimensional representation. These encoded features are then fused, with the resulting dimension depending on fusion type: concat yields $32n$, mean yields 32 (averaged encoder outputs), and tensor yields $(16 + 1)^n$ (Kronecker product with bias term), where $n$ is the number of modalities. All classifiers share a 2-layer MLP structure: Linear(fused_dim$\rightarrow$32) $\rightarrow$ ReLU $\rightarrow$ Dropout(0.3) $\rightarrow$ Linear(32$\rightarrow$2).

**AdaFuse.** We first pre-train baseline classifiers, then train the policy network while keeping classifiers frozen. We use AdamW with separate learning rates for policy ($3 \times 10^{-4}$) and encoders ($1 \times 10^{-5}$). Temperature is annealed linearly from $\tau_{\text{init}} = 1.5$ to $\tau_{\text{final}} = 0.3$ over 100 epochs; inference uses greedy decoding ($\tau \rightarrow 0$). Loss weights are $\lambda_{\text{ent}} = 0.1$ and $\lambda_{\text{sup}} = 0.3$. Reward weight is $\lambda_{\text{auc}} = 0.3$. We use balanced sampling with approximately 30% positive samples per batch for stable AUC reward computation.

**MoE Baseline.** The gating network is a 2-layer MLP (96$\rightarrow$64$\rightarrow$64$\rightarrow$15) that outputs soft weights over 15 frozen expert classifiers.

## Appendix B. Analysis of Learned Policy Behavior

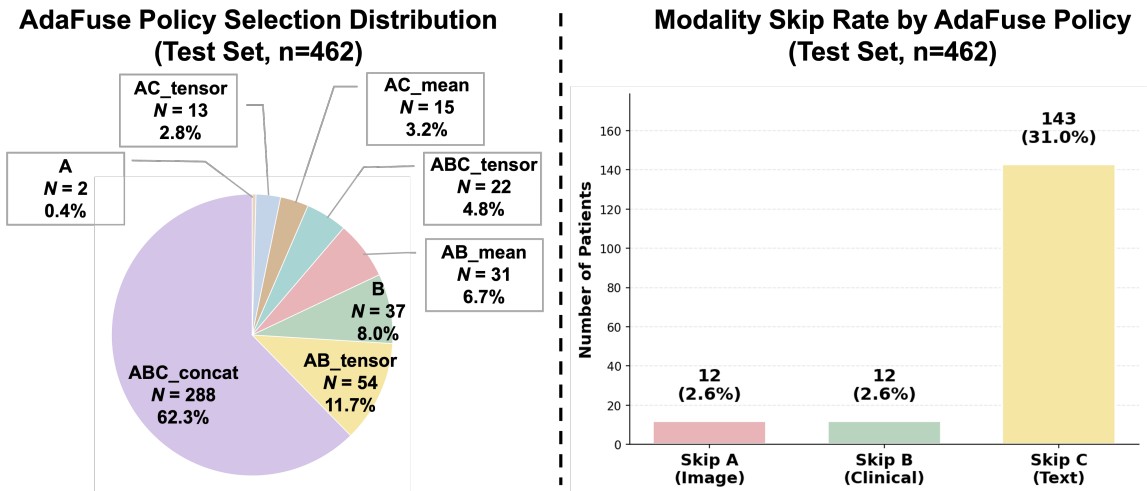

Figure 5: **AdaFuse policy selection distribution and modality skip rates on the test set.** Left: Distribution of modality-fusion combinations selected by the learned policy, where $N$ denotes the number of patients. The policy most frequently selects ABC-concat ($N = 288$, 62.3%), followed by AB-tensor ($N = 54$, 11.7%) and clinical-only ($N = 37$, 8.0%). Right: Frequency of skipping each modality. The text modality (C) is skipped for 143 patients (31.0%), while CT (A) and clinical variables (B) are each skipped for only 12 patients (2.6%). This confirms that the policy learns to filter out the less informative text modality while consistently relying on imaging and clinical data.

# Appendix C. Statistical Analysis of AUC Comparisons

Table 5: **Bootstrap confidence intervals and significance tests.** We report 95% confidence intervals from bootstrap analysis (1000 iterations) and p-values from DeLong's test comparing each method against AdaFuse. The wide confidence intervals ($\sim$0.20) across all methods reflect the limited number of positive cases (n=28) in the NLST test set, which is a shared constraint affecting all methods rather than a limitation specific to AdaFuse. Despite overlapping intervals among top-performing methods, AdaFuse achieves the highest point estimate (0.762) with the narrowest confidence interval (0.203) among competitive methods, suggesting that adaptive selection reduces prediction variance. AdaFuse significantly outperforms the text-only baseline (p=0.019).

| Method | AUC | 95% CI | p-value vs AdaFuse |
|---|---|---|---|
| **AdaFuse (Ours)** | **0.762** | [0.657, 0.860] | — |
| $ABC$-tensor | 0.759 | [0.646, 0.863] | 0.898 |
| $AB$-concat | 0.758 | [0.643, 0.867] | 0.901 |
| $AB$-mean | 0.755 | [0.640, 0.861] | 0.847 |
| DynMM | 0.754 | [0.640, 0.855] | 0.829 |
| $ABC$-mean | 0.748 | [0.631, 0.853] | 0.747 |
| $AC$-mean | 0.745 | [0.628, 0.849] | 0.399 |
| MoE | 0.742 | [0.628, 0.847] | 0.666 |
| $AC$-tensor | 0.739 | [0.618, 0.847] | 0.250 |
| $ABC$-concat | 0.735 | [0.622, 0.845] | 0.615 |
| $AB$-tensor | 0.735 | [0.617, 0.848] | 0.552 |
| $AC$-concat | 0.733 | [0.610, 0.843] | 0.148 |
| $A$ (CT) | 0.732 | [0.609, 0.842] | 0.116 |
| $BC$-tensor | 0.685 | [0.574, 0.792] | 0.191 |
| $BC$-mean | 0.678 | [0.566, 0.785] | 0.162 |
| $B$ (Clinical) | 0.662 | [0.544, 0.776] | 0.114 |
| $BC$-concat | 0.661 | [0.544, 0.771] | 0.107 |
| $C$ (Text) | 0.576 | [0.489, 0.657] | 0.019 |

## Appendix D. Data Examples

We provide two representative patient examples from the NLST dataset to illustrate the three modalities used in AdaFuse.

Table 6: **Summary of modalities and feature extraction.**

| Modality | Source | Feature Extractor | Dimension |
|---|---|---|---|
| A: CT Image | Low-dose chest CT | Sybil (pretrained) | 512D |
| B: Clinical | Structured demographics | PLCO2012 transform | 17D |
| C: Text | Generated risk report | CORe (pretrained) | 768D |

**Patient Case 1: Lung Cancer Positive.** 68-year-old male, BMI 27.46, Asian, diagnosed with lung cancer.

Table 7: **Clinical variables (Modality B) for Patient Case 1.** Raw values are transformed following the PLCO2012 model.

| Variable | Description | Raw Value | Transformed |
|---|---|---|---|
| age | Age at screening | 68 | 6.0 |
| race | Race (one-hot, 7 dims) | Asian | [0,0,0,0,1,0,0] |
| education | Education level | 3 | -1.0 |
| bmi | Body mass index | 27.46 | 0.46 |
| copd | COPD diagnosis | 0 | 0.0 |
| phist | Personal cancer history | 0 | 0.0 |
| fhist | Family lung cancer history | 0 | 0.0 |
| smo_status | Smoking status | Current | 0.0 |
| smo_intensity | Cigarettes per day | 30 | -0.069 |
| smo_duration | Years smoked | 58 | 31.0 |
| quit_time | Years since quitting | 0 | -10.0 |

**Generated Text Report (Modality C):** "The patient reports no significant occupational exposures. No significant chronic medical conditions reported. The patient is exposed to secondhand smoke at home and secondhand smoke at workplace."

**Patient Case 2: Lung Cancer Negative.** 65-year-old male, BMI 34.67, White, no lung cancer.

Table 8: **Clinical variables (Modality B) for Patient Case 2.** Raw values are transformed following the PLCO2012 model.

| Variable | Description | Raw Value | Transformed |
|---|---|---|---|
| age | Age at screening | 65 | 3.0 |
| race | Race (one-hot, 7 dims) | White | [0,1,0,0,0,0,0] |
| education | Education level | 3 | -1.0 |
| bmi | Body mass index | 34.67 | 7.67 |
| copd | COPD diagnosis | 0 | 0.0 |
| phist | Personal cancer history | 0 | 0.0 |
| fhist | Family lung cancer history | 0 | 0.0 |
| smo_status | Smoking status | Former | -1.0 |
| smo_intensity | Cigarettes per day | 40 | -0.152 |
| smo_duration | Years smoked | 41 | 14.0 |
| quit_time | Years since quitting | 10 | 0.0 |

**Generated Text Report (Modality C):** "The patient has occupational exposure to asbestos and agricultural dusts. Medical history is significant for pneumonia. The patient is exposed to secondhand smoke at home and secondhand smoke at workplace."

**Text Generation Process.** The synthetic text report is generated from 13 binary variables not included in Modality B: 6 occupational exposures (asbestos, chemicals, coal dust, agricultural dusts, firefighting smoke, welding fumes), 5 medical diagnoses (diabetes, heart disease, hypertension, pneumonia, stroke), and 2 environmental smoke exposures (home, workplace). This ensures that Modality C provides information complementary to Modality B rather than redundant.

