# OpenReview forum: "AdaFuse: Adaptive Multimodal Fusion for Lung Cancer Risk Prediction via Reinforcement Learning"
_MIDL.io/2026/Conference — MIDL 2026 Poster_

### Official Review · Reviewer_JDFX · 2025-12-18

**Confidence:** 5
**Preliminary Rating:** 1
**Final Rating:** 1

**Summary:**

This paper proposes an adaptive multimodal fusion framework, named AdaFuse, for lung cancer risk prediction. The framework leverages reinforcement learning to enable adaptive fusion of multimodal data, including CT images, clinical variables, and radiology reports.
Experimental results show that AdaFuse outperforms single-modality approaches and fixed-fusion strategies.

**Strengths:**

•  The paper introduces an innovative multimodal feature fusion strategy based on reinforcement learning.
•  The approach indicates the potential of reinforcement learning for personalized multimodal fusion in medical imaging tasks.

**Weaknesses:**

•  The evaluation is conducted on a single dataset, which limits the assessment of the model’s generalizability.
•  Although multimodal fusion is presented as the key contribution, the experimental comparison does not include state-of-the-art lung cancer risk prediction methods. Since the stated goal is to address lung cancer risk prediction, comparisons against established methods for this task are necessary.

**Detailed Comments:**

This work focuses on lung cancer risk prediction by proposing a novel adaptive fusion strategy for multimodal features. While the proposed method outperforms single-modality and fixed-fusion baselines, the improvement on the lung cancer risk prediction task itself appears limited.
Moreover, the experimental description lacks clarity regarding the reported AUC scores. It is unclear whether the AUC corresponds to a specific prediction horizon (e.g., Year 1, Year 2, etc.) or to an average prediction across multiple years (e.g., over a 6-year follow-up). Given the longitudinal nature of lung cancer risk prediction, this distinction is critical.
Additionally, restricting all experiments to a single dataset further limits confidence in the robustness and generalizability of the proposed approach.

**Justification Of Final Rating:**

The experimental comparison with state-of-the-art methods should be conducted based on the target task—in this case, lung cancer risk prediction—rather than on the specific methodological design, such as dual-modality fusion. The primary goal of proposing or exploring new approaches is to improve performance on the core task, not merely to demonstrate methodological novelty.

If a dual-modality fusion strategy does not outperform a simpler single-modality approach, increasing model complexity through fusion is not well justified. In such cases, additional analysis is needed to explain why the proposed design is beneficial. Without demonstrable performance gains or other clear advantages, an innovative architectural design does not meaningfully contribute to the task.

**Justification Of The Preliminary Rating:**

Due to the limited performance gains on the lung cancer risk prediction task itself, the lack of comparison with state-of-the-art methods, and the unclear experimental descriptions related to the target task, I recommend rejecting this paper.

**Questions To Address In The Rebuttal:**

•  Please clarify which prediction horizon(s) the reported AUC scores correspond to (e.g., Year 1, Year 2, or an averaged multi-year score).
•  What are the sources of the pretrained classifiers used in the framework?
•  Is any calibration method applied to the model outputs to obtain the reported prediction scores?

---

> ### Author Response · Authors · 2026-01-25
> **Responses to Reviewer JDFX (1/2):**
>
> We sincerely thank you for your thorough review and valuable feedback, which have greatly deepened our understanding and improved our work. In the following, we have provided a point-by-point response to all concerns raised.
>
> ---
> > W1: *The evaluation is conducted on a single dataset, which limits the assessment of the model’s generalizability.*
>
> R1: We thank the reviewer for this concern and provide the following clarifications:
> - **Why the evaluation is conducted on a single dataset?** No public lung cancer screening dataset provides CT images, clinical variables, and radiology reports simultaneously. Widely used datasets such as NLST, LIDC-IDRI[1], and LUNA16[2] lack either clinical variables or radiology reports.
> - **Regarding generalizability:** To address this concern, we conducted additional experiments on the **Vanderbilt Lung Screening Program (VLSP)**, a private external cohort with 858 patients (2.8% positive rate). VLSP contains only two modalities (CT images and clinical variables) as it lacks the comprehensive variables required to generate synthetic text reports.
> | Method | VLSP AUC |
> |--------|----------|
> | A (CT) | 0.771 |
> | B (Clinical) | 0.471 |
> | AB-concat | 0.725 |
> | AB-mean | 0.706 |
> | AB-tensor | 0.590 |
> | **AdaFuse** | **0.749** |
>
> CT features generalize well to VLSP (0.771 AUC), while clinical features show degraded performance (0.471 AUC). This negatively impacts all fixed dual-modality fusion methods (0.590–0.725). AdaFuse achieves 0.749 AUC, **higher than all dual-modality fusion baselines**, demonstrating its ability to adaptively filter less informative modalities under distribution shift.
> Constructing a public multimodal lung cancer dataset with CT images, clinical variables, and radiology reports is a valuable direction for future work.  We have added this external validation in `revised manuscript Section 3.5`.
>
>
>
>
>
>
> ---
> > W2: *Although multimodal fusion is presented as the key contribution, the experimental comparison does not include state-of-the-art lung cancer risk prediction methods. Since the stated goal is to address lung cancer risk prediction, comparisons against established methods for this task are necessary.*
>
> R2: We thank the reviewer for this comment and provide the following clarifications:
> - **Regarding lung cancer risk prediction:** State-of-the-art lung cancer risk prediction methods are predominantly **single-modality approaches**. Clinical-based models such as PLCOm2012 [3] focus on extracting predictive signals from demographic and smoking variables. Image-based models such as Sybil [4] focus on extracting predictive signals from CT scans. These methods address a different research question: **how to predict risk from a single modality.** Our work addresses a complementary question: **how to effectively combine multiple modalities.** We use Sybil features for CT (modality A) and PLCOm2012 transformation for clinical variables (modality B), building our adaptive fusion framework on top of these established feature extractors.
>
> - **Regarding multimodal fusion baselines:** We have added DynMM[5], a state-of-the-art adaptive fusion method using Gumbel-Softmax gating with parallel selection:
>
> | Method | Selection | AUC |
> |--------|-----------|-----|
> | AdaFuse | Sequential | 0.762 |
> | DynMM | Parallel | 0.754 |
> | MoE | Soft | 0.742 |
>
> AdaFuse achieves the highest AUC among all adaptive fusion methods. We have added DynMM to `revised manuscript Table 1`.
>
> ---
> > C1: *While the proposed method outperforms single-modality and fixed-fusion baselines, the improvement on the lung cancer risk prediction task itself appears limited.*
>
> We thank the reviewer for this comment. We provide two explanations:
> - The NLST test set we used contains only **28 positive cases** (~6% prevalence), which constrains the achievable AUC range. All top-performing methods cluster between 0.73–0.76.
> - The comparison baseline (ABC-tensor, 0.759) is already **the best among 15 exhaustively evaluated fusion configurations.**
>
> **The primary contribution of AdaFuse is the adaptive multimodal fusion framework, not AUC improvement alone**. AdaFuse provides practical advantages including:
> - Automatic discovery of effective modality combinations without exhaustive evaluation.
> - Lower computational cost by skipping uninformative modalities (35% fewer FLOPs than ABC-tensor)
> - Interpretable patient-specific modality selection decisions valuable in clinical settings.
>
> We have `revised Section 3.3` to highlight these practical advantages.

---

> ### Author Response · Authors · 2026-01-25
> **Responses to Reviewer JDFX (2/2):**
>
> > C2: *Moreover, the experimental description lacks clarity regarding the reported AUC scores. It is unclear whether the AUC corresponds to a specific prediction horizon (e.g., Year 1, Year 2, etc.) or to an average prediction across multiple years (e.g., over a 6-year follow-up). Given the longitudinal nature of lung cancer risk prediction, this distinction is critical.*
>
> We thank the reviewer for this important clarification request. All reported AUC scores correspond to **predicting lung cancer diagnosis at any time during the NLST follow-up period (up to 6 years)**, rather than a specific year or an average across years.
> Specifically, we use a binary label indicating whether a participant was diagnosed with lung cancer at any point during the study follow-up:
> | Label | Definition | Positive Cases | Prevalence |
> |-------|------------|----------------|------------|
> | lung_cancer | Diagnosed at any time during f/u | 147 / 2,309 | 6.37% |
>
> This formulation is chosen for two reasons:
> - In lung cancer screening, a key objective is to **identify individuals at elevated risk who require continued monitoring.**
> - The cumulative label (6.37% prevalence) provides **more positive samples for stable model training** compared to year-specific labels (1.78%–5.85%).
>
> We have added this clarification in `revised manuscript Section 3.1`.
>
> ---
> > C3: *Additionally, restricting all experiments to a single dataset further limits confidence in the robustness and generalizability of the proposed approach.*
>
> Please see our response to W1, where we provide explanations for why evaluation is conducted on a single dataset and present external validation results on the VLSP dataset.
>
> ---
> > Q1: *Please clarify which prediction horizon(s) the reported AUC scores correspond to (e.g., Year 1, Year 2, or an averaged multi-year score).*
>
>
> A1: Please see our response to C2, where we clarify that all reported AUC scores correspond to predicting lung cancer diagnosis at any time during the NLST follow-up period (up to 6 years).
>
> ---
> > Q2: *What are the sources of the pretrained classifiers used in the framework?*
>
>
> A2: We thank the reviewer for this question and provide the following clarifications:
> - The 15 fusion classifiers are **trained by us on our training split** (1,847 patients), not sourced from external pretrained models.
> - Each classifier is a **lightweight 2-layer MLP** (~2K parameters) that takes encoded or fused features as input and outputs cancer prediction.
> - During AdaFuse training, these classifiers are **frozen to provide stable reward signals** for policy learning.
>
> We have added classifier architecture and training details in `revised manuscript Appendix A.`
>
> ---
> > Q3: *Is any calibration method applied to the model outputs to obtain the reported prediction scores?*
>
> We thank the reviewer for this question.
> - **No calibration method is applied to the model outputs.** The reported prediction scores are obtained directly from the softmax layer of the classifier.
> - Our primary evaluation metric is **AUC, which measures ranking performance and is invariant to monotonic transformations** of the prediction scores. Calibration does not affect AUC.
> - For future clinical deployment where well-calibrated risk probabilities are needed, calibration methods could be applied as a post-processing step.
>
> ---
> **Reference**
>
> [1] The Lung Image Database Consortium (LIDC) and Image Database Resource Initiative (IDRI): A Completed Reference Database of Lung Nodules on CT Scans (Medical Physics, 2011)
>
> [2] Validation, comparison, and combination of algorithms for automatic detection of pulmonary nodules in computed tomography images: The LUNA16 challenge (Medical Image Analysis, 2017)
>
>
> [3] Selection criteria for lung-cancer screening. (New England Journal of Medicine, 2013)
>
> [4] Sybil: a validated deep learning model to predict future lung cancer risk from a single low-dose chest computed tomography (Journal of Clinical Oncology, 2023)
>
> [5] Dynamic Multimodal Fusion. (arXiv:2204.00102)

---

### Official Review · Reviewer_gyAM · 2026-01-08

**Confidence:** 4
**Preliminary Rating:** 4
**Final Rating:** 4

**Summary:**

Proposed AdaFuse is a new multimodal fusion model for lung cancer risk prediction as a sequential decision problem. It is a novel idea in the field about how the policy observes a state and decides whether to add a modality or stop and make predictions. The key significance is that this paper demonstrates patient-specific when to use how many modality paradigms that can skip modalities when they are unhelpful, rather than always processing all inputs.

**Strengths:**

- This work pointed out that the current framework often includes all the modalities in predictions whether they are helpful or not. This work suggests that skipping some modality for some cases improves the results.
- This work is very well organized and written on paper.
- Solid experimental coverage for fair comparison with traditional multi-modal training set ups.

**Weaknesses:**

- The author clearly stated, but they used generated reports as one of their modalities. I still think this is a big weakness of this work, especially the model skipped this modality most according to Appendix A.
- Author argues that the proposed AdaFuse achieved the best AUC, but the difference is very minimal (0.762 vs. 0.759). Maybe add more explanation or justification on this part?
- Information about these classifiers is lacking. Not very clear how this classifier looks like. Also, the author mentioned that this requires classifiers for all modality fusion combinations. This can be one of the limitations, especially if this modal is applied with bigger modalities.
- One of the motivations of this AdaFuse is avoiding unnecessary computation. It would be great to see a comparison of computations between each model.

**Detailed Comments:**

- Maybe a little more training details in appendix sections?
- Also, I would like to see data examples in the appendix too. Especially for clinical data variables.

**Justification Of Final Rating:**

Thank you for authors to deliver my concerns. I will keep my current positive rating. I think it has good novelty and good potential, but still results are not perfectly support the claim. I still recommend for MIDL

**Justification Of The Preliminary Rating:**

This paper presents a clean and well organized formulation of patient specific modality selection using a RL. Nice mix of multi-modal training with an up and coming RL framework. Novel ideas and the results look promising too. With little modification, it can be a great paper.

**Questions To Address In The Rebuttal:**

Please address the things mentioned in above sections.

---

> ### Author Response · Authors · 2026-01-25
> **Responses to Reviewer gyAM (1/2):**
>
> We would like to thank you for the positive feedback and for recognizing the strengths of our paper, including that "This work is very well organized and written", and "solid experimental coverage for fair comparison…". In the following, we provide point-by-point responses to all your concerns.
>
> ---
> > W1: *The author clearly stated, but they used generated reports as one of their modalities. I still think this is a big weakness of this work, especially the model skipped this modality most according to Appendix A.*
>
> R1: We acknowledge this limitation and provide the following clarifications:
> - **Regarding the use of synthetic reports:** We used synthetic reports because no public lung cancer screening dataset provides CT images, clinical variables, and radiology reports simultaneously. Widely used datasets include:
>   - **NLST:** CT images + clinical variables, no radiology reports
>   - **LIDC-IDRI:**[1] CT images + nodule annotations, no clinical variables or radiology reports
>   - **LUNA16:**[2] CT images + nodule annotations, no clinical variables or radiology reports
>
> - **Regarding the model skipping text modality:** The fact that AdaFuse learns to skip the text modality for 31% of patients **demonstrates the method's ability to automatically identify and filter low-information modalities.** The text-only baseline achieves only 0.576 AUC (near random), confirming that these synthetic reports provide limited predictive value for lung cancer risk. AdaFuse discovers this automatically without prior knowledge.
>
> We agree that evaluation with real radiology reports would strengthen the work. Constructing a multimodal lung cancer dataset with CT images, clinical variables, and radiology reports is a valuable direction for future research.
>
> ---
> > W2: *Author argues that the proposed AdaFuse achieved the best AUC, but the difference is very minimal (0.762 vs. 0.759). Maybe add more explanation or justification on this part?*
>
> R2: We thank the reviewer for this comment. We provide two explanations for the minimal difference:
> - The NLST test set contains only **28 positive cases** (~6% prevalence), which constrains the achievable AUC range. All top-performing methods cluster between 0.73-0.76.
> - The comparison baseline (ABC-tensor, 0.759) is already **the best among 15 exhaustively evaluated fusion configurations**.
> Despite the minimal AUC difference, AdaFuse provides two practical advantages:
> - AdaFuse **automatically discovers effective modality combinations** without exhaustive evaluation of all 15 configurations.
> - AdaFuse **uses 35% fewer FLOPs** (1.164 vs 1.790 MFLOPs) by skipping uninformative modalities for individual patients. Please see our response to W4 for detailed comparison.
> We have `revised Section 3.3` to highlight these practical advantages beyond AUC.
>
> ---
> > W3: *Information about these classifiers is lacking. Not very clear how this classifier looks like. Also, the author mentioned that this requires classifiers for all modality fusion combinations. This can be one of the limitations, especially if this modal is applied with bigger modalities.*
>
> R3: We thank the reviewer for this concern.
> - **Regarding classifier architecture:** Each classifier is a 2-layer MLP: Linear(input_dim→32) → ReLU → Dropout(0.3) → Linear(32→2). Input dimensions vary by fusion type: concat uses 32×num_modalities, mean uses 32 (averaged encoder outputs), and tensor uses the Kronecker product dimension ((16+1)^num_modalities, where 16 is the tensor projection dimension and +1 accounts for the bias term).
> - **Regarding scalability:** We acknowledge this as a limitation. For n modalities with k fusion types, O(k×2^n) classifiers are needed. However, we note that:
>   - Encoders are shared across all classifiers, which is the main parameter cost.
>   - Classifiers are lightweight (~2K parameters each)
>   - In practice, domain knowledge limits meaningful combinations. Future work could explore hypernetworks or attention-based fusion to generate classifier weights dynamically.
> We have added classifier architecture details in `revised manuscript Appendix A`.

---

> ### Author Response · Authors · 2026-01-25
> **Responses to Reviewer gyAM (2/2):**
>
> ---
> > W4: *One of the motivations of this AdaFuse is avoiding unnecessary computation. It would be great to see a comparison of computations between each model.*
>
> R4: We thank the reviewer for this suggestion. We computed MFLOPs (million floating-point operations) for all methods:
> | Method           | MFLOPs |
> |------------------|--------|
> | A (CT)           | 0.543  |
> | B (Clinical)     | 0.017  |
> | C (Text)         | 1.067  |
> | AB-concat        | 0.559  |
> | AB-mean          | 0.557  |
> | AB-tensor        | 0.433  |
> | AC-concat        | 1.610  |
> | AC-mean          | 1.608  |
> | AC-tensor        | 1.477  |
> | BC-concat        | 1.084  |
> | BC-mean          | 1.082  |
> | BC-tensor        | 1.088  |
> | ABC-concat       | 1.626  |
> | ABC-mean         | 1.622  |
> | ABC-tensor       | 1.790  |
> | MoE              | 3.492  |
> | DynMM            | 1.635  |
> | **AdaFuse**      | **1.164** |
>
> Among adaptive methods, AdaFuse (1.164 MFLOPs) uses **29% fewer FLOPs than DynMM** (1.635 MFLOPs) and **67% fewer FLOPs than MoE** (3.492 MFLOPs). AdaFuse also uses **fewer FLOPs than all triple-modality fixed fusion methods (1.622–1.790 MFLOPs)**, demonstrating that adaptive modality selection reduces computational cost by skipping uninformative modalities for individual patients.
> We have added MFLOPs to `revised manuscript Table 1`.
>
> ---
> > C1: *Maybe a little more training details in appendix sections?*
>
> We thank the reviewer for this suggestion and have added comprehensive implementation details in the `revised manuscript Appendix A`.
>
> ---
> > C2: *Also, I would like to see data examples in the appendix too. Especially for clinical data variables.*
>
> We thank the reviewer for this suggestion and have added data examples in `revised manuscript Appendix D`, including two representative patient cases (one positive, one negative) with detailed clinical variables (raw values and PLCO2012 transformations), generated text reports, and feature extraction summary.
>
> ---
> **Reference**
>
> [1] The Lung Image Database Consortium (LIDC) and Image Database Resource Initiative (IDRI): A Completed Reference Database of Lung Nodules on CT Scans (Medical Physics, 2011)
>
> [2] Validation, comparison, and combination of algorithms for automatic detection of pulmonary nodules in computed tomography images: The LUNA16 challenge (Medical Image Analysis, 2017)

---

### Official Review · Reviewer_94Jo · 2026-01-16

**Confidence:** 4
**Preliminary Rating:** 3
**Final Rating:** 4

**Summary:**

- AdaFuse proposes a method that uses RL to decide which modalities to include when making lung cancer predictions.
- Instead of always fusing CT, clinical and text data together, policy network sequentially picks which combination to use for each patient.
- They test it on the NLST test set and get (0.762 vs 0.759 AUC) against the best fixed fusion. Their policy learns to skip the text modality 1/3rd of the times (perhaps due to synthetic reports which are not radiology reports per se).

**Strengths:**

- The idea is simple ("should we even use this modality?") and reasonable. Most fusion papers assume more data is better. This paper challenges that.
- The correlation analysis in figure 3 is nice. It shows text predictions are uncorrelated with CT-ones (which explains why the policy skips text).
- Prior works on RL for modality weighting exists but that work learned continuous weights and not discrete selection. The sequential MDP formulation here is a reasonable extension.

**Weaknesses:**

- Text is generated from clinical variables. So modality C is literally derived from modality B. It contains no independent information. The whole premise of adaptive selection only makes sense when modalities have genuinely different content.
- The performance gap is 0.003. No confidence intervals, no significance tests. Without this, the claim that AdaFuse "outperforms" fixed fusion is not supported.
- The experimental setup favors CT heavily. They use features from Sybil (trained specifically on NLST). Clinical and text encoders have no such task-specific pretraining (and this confounds the interpretation of modality importance).
- Looking at Figure 5, ~60% of patients get ABC-concat anyway. And much of the remaining adaptive behaviour is just dropping text (which is already redundant).
- The sequential MDP isn't justified. Why not make selection decisions in parallel? Dynamic Multimodal Fusion (ECCV 2023) uses gating to select uni/multimodal models and would be a more direct comparison than the basic MoE baseline.

**Detailed Comments:**

- The MoE baseline gets 0.742 AUC, which is worse than many fixed fusion methods. MoE implementation may be weak. What gating architecture was used? How many experts?
- Temperature schedule for action sampling not specified.

**Justification Of Final Rating:**

Method works as claimed. Modest but valuable contribution. AUC gains not statistically significant, though partly due to limited positives (n=28). Computational savings (29-67% fewer FLOPs) and automatic combination discovery are practical benefits. Synthetic text limits conclusions but is a data constraint, not methodological flaw. Interesting use of RL for adaptive fusion. Worth showcasing to the medical imaging community.

**Justification Of The Preliminary Rating:**

The paper asks a very interesting question. But the core experimental result doesn't convince me. The AUC improvement is tiny and not shown to be statistically significant. More importantly, the synthetic text modality (from clinical variables) undermines the entire premise.

**Questions To Address In The Rebuttal:**

- Please provide confidence intervals and statistical significance for the AUC comparison. With ~28 (+)-ive test cases, is the 0.003 improvement reliable?
- What happens with single-step parallel selection instead of sequential MDP? Does sequential actually help?
- Can you compare against DynMM or other gating-based adaptive methods rather than just basic MoE?

---

> ### Author Response · Authors · 2026-01-25
> **Responses to Reviewer 94Jo (1/3)**
>
> We would like to thank you for recognizing the strengths of our paper, including that “The idea is simple… and reasonable.", "Most fusion papers assume more data is better, this paper challenges that.", “ The correlation analysis in figure 3 is nice.”, “The sequential MDP formulation here is a reasonable extension.” In the following, we provide point-by-point responses to all your concerns.
>
> ---
>
> > W1: *Text is generated from clinical variables. So modality C is literally derived from modality B. It contains no independent information. The whole premise of adaptive selection only makes sense when modalities have genuinely different content.*
>
> R1: ​​We thank the reviewer for this important question. We respectfully clarify that **modality B and modality C are derived from entirely disjoint sets of clinical variables:**
> - **Modality B (Clinical Variables)** contains the 11 risk factors from the PLCOm2012 model[1], encoded as 17 dimensions (due to one-hot encoding of categorical variables such as race/ethnicity):
>   - Demographics: age, race/ethnicity, education level, BMI
>   - Medical history: COPD, personal cancer history, family history of lung cancer
>   - Smoking variables: status, intensity, duration, quit-years
> - **Modality C (Text Reports)** is generated from a **completely separate** set of 13 variables **not included in PLCOm2012**:
>   - Occupational exposures (6 variables): asbestos, chemicals/plastics manufacturing, coal mining, farming, fire fighting, welding
>   - Medical comorbidities (5 variables): diabetes, heart disease, hypertension, pneumonia, stroke
>   - Secondhand smoke exposure (2 variables): living with smokers, workplace exposure to smokers
> - **No information overlap exists** between the two modalities. The text modality captures risk factors orthogonal to PLCOm2012, representing complementary clinical context that the established risk model does not incorporate.
>
> ---
> > W2: *The performance gap is 0.003. No confidence intervals, no significance tests. Without this, the claim that AdaFuse "outperforms" fixed fusion is not supported.*
>
> R2: We thank the reviewer for this suggestion. We conducted bootstrap analysis (1000 iterations) with DeLong's test:
> | Method       | AUC   | 95% CI          | p-value vs AdaFuse |
> |--------------|-------|-----------------|-------------------|
> | AdaFuse      | 0.762 | [0.657, 0.860]  | —                 |
> | ABC-tensor   | 0.759 | [0.646, 0.863]  | 0.898             |
> | AB-concat    | 0.758 | [0.643, 0.867]  | 0.901             |
> | AB-mean      | 0.755 | [0.640, 0.861]  | 0.847             |
> | DynMM        | 0.754 | [0.640, 0.855]  | 0.829             |
> | ABC-mean     | 0.748 | [0.631, 0.853]  | 0.747             |
> | AC-mean      | 0.745 | [0.628, 0.849]  | 0.399             |
> | MoE          | 0.742 | [0.628, 0.847]  | 0.666             |
> | AC-tensor    | 0.739 | [0.618, 0.847]  | 0.250             |
> | ABC-concat   | 0.735 | [0.622, 0.845]  | 0.615             |
> | AB-tensor    | 0.735 | [0.617, 0.848]  | 0.552             |
> | AC-concat    | 0.733 | [0.610, 0.843]  | 0.148             |
> | A-only       | 0.732 | [0.609, 0.842]  | 0.116             |
> | BC-tensor    | 0.685 | [0.574, 0.792]  | 0.191             |
> | BC-mean      | 0.678 | [0.566, 0.785]  | 0.162             |
> | B-only       | 0.662 | [0.544, 0.776]  | 0.114             |
> | BC-concat    | 0.661 | [0.544, 0.771]  | 0.107             |
> | C-only       | 0.576 | [0.489, 0.657]  | 0.019             |
>
> The wide confidence intervals (~0.20) across all methods reflect the limited number of positive cases (n=28) in the NLST test set, which is a shared constraint affecting all methods rather than a limitation specific to our approach. Based on this analysis, we have `revised Section 3.3` to remove *outperform* claims and instead present objective AUC comparisons. We also highlight two additional benefits of AdaFuse:
> - AdaFuse exhibits **the most stable performance among competitive methods**, with the narrowest confidence interval (0.203) compared to ABC-tensor (0.218) and AB-concat (0.224). This suggests that adaptive selection reduces prediction variance by avoiding suboptimal fusion choices for individual patients.
> - AdaFuse **automatically discovers effective modality combinations** without exhaustively evaluating all 15 fusion configurations, while providing **interpretable, patient-specific selection decisions** that are valuable in clinical settings.
> We will include this statistical analysis in the `revised manuscript Appendix C`.

---

> > ### Comment · Reviewer_94Jo · 2026-02-01
> > **Response to Rebuttal**
> >
> > I appreciate the authors' response.
> >
> > - The B/C clarification is factually correct - the variable sets are disjoint. But this misses the point. The synthetic text contains no imaging-derived information - it's structured metadata in sentence form, not actual radiology observations. Real reports describe nodules, lung findings, clinical impressions. The 0.576 AUC confirms this. Adaptive selection here is learning to ignore a weak signal, not routing between complementary modalities.
> > - Statistical analysis confirms that the improvement is indistinguishable from noise. The authors have agreed to dropping performance claims and pivoting to computational efficiency and interpretability arguments.
> > - External validation helps but is limited - VLSP has only 2 modalities, so it doesn't test the full 3-modality selection framework.
> > - The DynMM comparison strengthens the paper. Sequential outperforming parallel is a reasonable empirical finding.

---

> ### Author Response · Authors · 2026-01-25
> **Responses to Reviewer 94Jo (2/3)**
>
> > W3: *The experimental setup favors CT heavily. They use features from Sybil (trained specifically on NLST). Clinical and text encoders have no such task-specific pretraining (and this confounds the interpretation of modality importance).*
>
> R3: We thank the reviewer for this observation. We have made efforts to ensure fair comparison across modalities:
> - **For CT images**, we use Sybil to extract high-quality representations from complex spatial data, and we evaluate on the held-out test set from Ardila et al [2]. that was explicitly excluded from Sybil's training to avoid data leakage.
> - **For clinical variables**, task-specific pretraining is neither standard nor necessary for low-dimensional tabular data (17 features). PLCOm2012 [1], Mayo [3], and Brock [4] are representative and widely-adopted lung cancer risk prediction models that use logistic regression directly on clinical variables without any pretraining. Our design follows this established practice.
> - **For text reports**, we use CORe [5], a BERT-based model pretrained on chest radiograph reports, which already provides domain-specific medical text representations. To our knowledge, no task-specific text encoder exists for lung cancer risk prediction from radiology reports. Moreover, since our text reports are synthetically generated, the purpose of this modality is to evaluate whether AdaFuse can effectively identify and filter less informative inputs, which it successfully demonstrates by skipping text for 31% of patients.
>
> ---
> > W4: *Looking at Figure 5, ~60% of patients get ABC-concat anyway. And much of the remaining adaptive behaviour is just dropping text (which is already redundant).*
>
> R4: We thank the reviewer for this observation.
> - Regarding *"~60% of patients get ABC-concat anyway"*: **The 60% ABC-concat selection is a learned behavior, not a predetermined one.** The policy network could have converged to any of the 15 possible modality-fusion combinations, yet it learned that ABC-concat is optimal for the majority of patients. The remaining 40% of patients receive diverse treatments, including AB-tensor (11.7%), clinical-only (8.0%), and AB-mean (6.7%), demonstrating non-trivial patient-specific adaptation.
> - Regarding *"much of the remaining adaptive behaviour is just dropping text"*: **The key distinction is that AdaFuse performs learned, patient-specific selection rather than random dropout.** While modality dropout techniques exist for robustness to missing data, they apply random masking during training. In contrast, AdaFuse learns from reward signals which modality combinations benefit specific patients, enabling it to selectively filter uninformative modalities at inference time.
> We acknowledge that the current NLST setup with synthetic text may not fully showcase the potential of adaptive selection. We expect more diverse selection patterns when applied to datasets with multiple informative modalities, and we list this as future work.
>
> ---
> > W5: *The sequential MDP isn't justified. Why not make selection decisions in parallel? Dynamic Multimodal Fusion (ECCV 2023) uses gating to select uni/multimodal models and would be a more direct comparison than the basic MoE baseline.*
>
> R5: We thank the reviewer for this suggestion. We implemented DynMM[6] using Gumbel-Softmax gating with the same modality encoders and fusion classifiers as AdaFuse:
>
> | Method  | Selection  | AUC   |
> |---------|------------|-------|
> | AdaFuse | Sequential | 0.762 |
> | DynMM   | Parallel   | 0.754 |
> | MoE     | Soft       | 0.742 |
> - **Why not make selection decision in parallel?** We chose sequential over parallel selection because it **better mirrors clinical reasoning**, where physicians interpret initial test results before deciding whether to order additional diagnostics. Sequential formulation allows each decision to be **conditioned on previously observed modalities**, whereas parallel gating must commit to all decisions simultaneously based solely on the initial input features. Empirically, sequential selection achieves higher AUC (0.762) than parallel (DynMM, 0.754) and soft (MoE, 0.742) alternatives on our task.
>
> We have added DynMM to the `revised manuscript Table 1` and include this discussion in the `revised manuscript Section 3.3`.

---

> ### Author Response · Authors · 2026-01-25
> **Responses to Reviewer 94Jo (3/3)**
>
> > C1: *The MoE baseline gets 0.742 AUC, which is worse than many fixed fusion methods. MoE implementation may be weak. What gating architecture was used? How many experts?*
>
> We thank the reviewer for this question and provide our MoE implementation details as follows:
> - **Regarding gating architecture:** We use a 2-layer MLP with the same architecture as AdaFuse's policy network for fair comparison. Input is 96 dimensions (concatenated 32-dim encoded features from all 3 modalities). Hidden layers consist of Linear(96→64) + ReLU + Dropout(0.2) + Linear(64→64) + ReLU. Output is Linear(64→15) + Softmax → expert weights.
> - **Regarding number of experts:** We use 15 experts (3 single-modality + 9 dual-modality + 3 triple-modality), covering all valid modality-fusion combinations. Expert models are pre-trained on respective modality combinations, then frozen during gating training.
> - **Regarding the lower AUC:** The soft MoE's lower performance is not due to weak implementation, but due to an inherent limitation of soft weighting. With 15 experts, weights are spread across all experts including poorly performing ones (e.g., C-only: 0.576, BC-concat: 0.661), which drags down overall performance. In contrast, AdaFuse makes discrete selection decisions that entirely exclude uninformative modality combinations.
> We have added more details of MoE baseline in the `revised manuscript Section 3.2`.
>
> ---
> > C2: *Temperature schedule for action sampling not specified.*
>
> We thank the reviewer for pointing this out. The temperature schedule for action sampling:
> - **Training:** Linear annealing from τ=1.5 to τ=0.3 over 100 epochs. High initial temperature encourages exploration of diverse modality combinations, while gradual decrease focuses learning on refining the best strategies.
> - **Inference:** Greedy decoding (τ→0), ensuring deterministic and reproducible results.
> We have added comprehensive training details including temperature schedule in `revised manuscript Appendix A`.
>
> ---
> > Q1: *Please provide confidence intervals and statistical significance for the AUC comparison. With ~28 (+)-ive test cases, is the 0.003 improvement reliable?*
>
> A1: Please see our response to W2, where we provide **bootstrap confidence intervals and DeLong's test results** for all methods. The wide confidence intervals (~0.20) reflect the limited positive cases (n=28), which is a **shared constraint affecting all methods**. Based on this analysis, we have revised our claims from "outperforms" to objective AUC comparisons. Beyond AUC, AdaFuse provides **interpretable patient-specific modality selection** and **automatically discovers effective modality combinations** without exhaustively evaluating all configurations.
>
> ---
> > Q2: *What happens with single-step parallel selection instead of sequential MDP? Does sequential actually help?*
>
> A2: Please see our response to W5. We implemented DynMM (parallel Gumbel-Softmax gating) for comparison: AdaFuse (0.762) vs DynMM (0.754). We chose sequential over parallel selection because it **better mirrors clinical reasoning,** where physicians interpret initial test results before deciding whether to order additional diagnostics. In AdaFuse, each step's state representation includes the encoded features of previously selected modalities (`Eq. 1`), allowing **subsequent decisions to be conditioned on observed information**. Parallel gating must commit to all decisions simultaneously based solely on initial input features.
>
> ---
> > Q3: *Can you compare against DynMM or other gating-based adaptive methods rather than just basic MoE?*
>
> A3: Please see our response to W5, where we implemented DynMM using Gumbel-Softmax gating. DynMM results have been added to `revised manuscript Table 1` with discussion in `revised manuscript Section 3.3`.
>
> ---
> **Reference**
>
> [1]  Selection criteria for lung-cancer screening. (New England Journal of Medicine, 2013)
>
> [2]  End-to-end lung cancer screening with three-dimensional deep learning on low-dose chest computed tomography. (Nature medicine, 2019)
>
> [3] Probability of Cancer in Pulmonary Nodules Detected on First Screening CT. (New England Journal of Medicine, 2013)
>
> [4] The probability of malignancy in solitary pulmonary nodules. Application to small radiologically indeterminate nodules. (Arch Intern Med. 1997)
>
> [5] CoRe: An Efficient Coarse-refined Training Framework for BERT. (arXiv:2011.13633)
>
> [6]  Dynamic Multimodal Fusion. (arXiv:2204.00102)

---

### Author Rebuttal · Authors · 2026-01-25

**Rebuttal:**

Dear Reviewers and Area Chairs,

We sincerely thank all reviewers for their constructive feedback. Below we summarize the key concerns and our responses. All new content in the revised manuscript is highlighted in blue.

- ***Regarding statistical significance (Reviewers 94Jo, gyAM)***: We conducted bootstrap analysis (1,000 iterations) with DeLong's test. The wide confidence intervals (~0.20) reflect the limited positive cases (n=28), a shared constraint affecting all methods. We revised Section 3.3 to present objective comparisons and highlight practical advantages. See `revised Appendix C`.

- ***Regarding generalizability (Reviewer JDFX)***: We conducted external validation on **VLSP**, a private cohort with 858 patients. AdaFuse achieves 0.749 AUC, higher than all fixed fusion baselines (0.590 to 0.725). See `revised Section 3.5`.

- ***Regarding adaptive fusion baselines (Reviewer 94Jo)***: We implemented **DynMM** (CVPR workshop 2023) with Gumbel-Softmax gating. AdaFuse (0.762) achieves higher AUC than DynMM (0.754) and MoE (0.742). See `revised Table 1`.

- ***Regarding computational cost (Reviewer gyAM)***: AdaFuse (1.164 MFLOPs) uses 29% fewer FLOPs than DynMM and 67% fewer than MoE. See `revised Table 1`.

- ***Regarding modality independence (Reviewer 94Jo)***: Modality B and C derive from **entirely disjoint variable sets**: 11 PLCOm2012 factors vs. 13 separate variables.

- ***Regarding implementation details (Reviewers 94Jo, gyAM, JDFX)***: Added classifier architecture, temperature schedule in `revised Appendix A`, data examples in `revised Appendix D`, and clarified 6-year cumulative prediction task in `revised Section 3.1`.


| Revision | Addresses |
|----------|-----------|
| Bootstrap CI + DeLong's test (`Appendix C`) | 94Jo-W2/Q1 |
| DynMM baseline (`Table 1, Sec 3.3`) | 94Jo-W5/Q2/Q3, JDFX-W2 |
| External validation on VLSP (`Sec 3.5`) | JDFX-W1/C3 |
| MFLOPs comparison (`Table 1`) | gyAM-W4 |
| Implementation details (`Appendix A/D`) | 94Jo-C1/C2, gyAM-W3/C1/C2, JDFX-Q2/Q3 |



We believe our revisions have addressed all major concerns.

Best Regards,

The AdaFuse Team

**Supporting Material:**

/attachment/82931a0bfdb2de295b19cdaa81e1b468f44416c6.pdf

---

### Comment · Area_Chair_hPFx · 2026-01-27
**Discussion Period**

Dear Reviewers,​

Thanks for your time and effort in reviewing this paper. This is the right time to discuss this paper with each other.​

The authors have provided a rebuttal to your comments and uploaded a revision. Please review their responses and the revised manuscript. For the preliminary recommendation, we have mixed opinions with a borderline, a weak accept, and a strong reject.​ Considering the authors' responses and the discussion, please update your rating and assessments for the paper.

Any discussion is welcome, and you may consider reading each other's reviews, posting questions for clarification, and reaching a consensus.​

AC

---

### Meta-Review · Area_Chair_hPFx · 2026-02-10

**Recommendation:** Accept (Poster)
**Confidence:** 4

**Metareview:**

The paper presents a modality selection strategy leveraging reinforcement learning in a multimodal fusion framework for lung cancer risk prediction. The reviewers found the proposed multimodal fusion strategy simple and innovative. There were mixed reviews, with two weak accepts and a strong reject. While the reviewers appreciated the novelty and potential of the work, there remain some concerns including modest contributions, not perfectly supported claims, lacking detailed analyses demonstrating the true benefits, etc. The research question addressed in this work is really interesting, deviating from the usual multimodal fusion setup of processing all available modalities for each patient. I think the authors did a great job in their rebuttal addressing the reviewers' concerns and revising the manuscript. Despite having relatively weaker evaluation and justification of all the claims, I lean toward acceptance. I think that this would make a good conference contribution inciting some interesting discussions, particularly along the patient-specific modality selection direction. I strongly suggest the authors share their complete code and possibly trained models to their GitHub repo.

---

### Decision · Program_Chairs · 2026-02-13

Accept (Poster)